# NFT1000: A Cross-Modal Dataset For Non-Fungible Token Retrieval

## ABSTRACT

With the rise of "Metaverse" and "Web 3.0", Non-Fungible Token (NFT) has emerged as a kind of pivotal digital asset, garnering significant attention. By the end of March 2024, more than 1.7 billion NFTs have been minted across various blockchain platforms. To effectively locate a desired NFT, conducting searches within a vast array of NFTs is essential. The challenge in NFT retrieval is heightened due to the high degree of similarity among different NFTs, regarding regional and semantic aspects. In this paper, we will introduce a benchmark dataset named "NFT Top1000 Visual-Text Dataset" (NFT1000, as shown in Fig.1), containing 7.56 million image-text pairs, and being collected from 1000 most famous PFP [1] NFT collections[2] by sales volume on the Ethereum blockchain. Based on this dataset and leveraging the CLIP series of pre-trained models as our foundation, we propose the dynamic masking fine-tuning scheme. This innovative approach results in a 7.4% improvement in the top1 accuracy rate, while utilizing merely 13% of the total training data (0.79 million *vs.* 6.1 million). We also propose a robust metric Comprehensive Variance Index (CVI) to assess the similarity and retrieval difficulty of visual-text pairs data. Please try our demo through the anonymous link at https://876p9s4054.vicp.fun/

## CCS CONCEPTS

• **Computing methodologies** → *Image representations*.

## KEYWORDS

Cross-Modal Retrieval, Blockchain, NFT, Recommendation, CLIP

## 1 INTRODUCTION

With the emerging concept of the "Metaverse" [17, 27] and "Web3.0" [14], NFT [26] has entered the public eye as a significant digital asset within this space. The NFT, standing for Non-Fungible Token, is a unique cryptocurrency token on blockchain [29] representing digital assets such as images, videos, tickets, inscription, etc. NFT is coveted for its characteristics of provenance, high liquidity, and rarity. NFT possesses immense value; for instance, the renowned NFT project CryptoPunks has amassed a trading volume of $2.78

---

[1]PFP is an abbreviation for "Profile Picture", representing a category of NFTs primarily used as avatars in social media contexts.

[2]An NFT collection represents an NFT project, which contains the same batch of media files and metadata data.

---

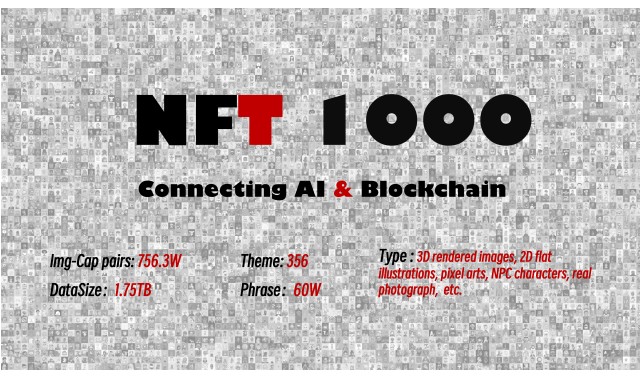

**Figure 1: NFT1000 is the first NFT dataset within the field of computer vision. The proposed dataset encompasses the most renowned 1,000 avatar-based NFT projects on the Ethereum mainnet, comprising 7.563 million image-text pairs.**

billion since its launch[3]. Statistical data[4] indicates that by the end of March 2024, the cumulative number of NFT minted on different blockchain platforms has exceeded 1.7 billion. When purchasing NFTs, people often gravitate towards tokens that align with their personal style or match their preferences, aiming to fulfill their desire for personalized expression in virtual spaces. However, both the academic and industrial sectors lack effective and precise methods or toolkits for the retrieval of NFT data due to the high degree of regional and semantic similarity among NFTs (Fig.2). This represents a novel research area that requires our exploration.

Given the lack of a dedicated NFT dataset for scientific research in the field of computer vision, we firstly construct the NFT1000. It is composed of the top 1000 PFP NFT collections by sales volume on the Ethereum blockchain with the ERC-721 [5] standard. Each project contains an average of 7500 image-text pairs. In total, the dataset includes 7.56 million image-text pairs, with a data volume of 1.75TB. It is suitable for various downstream tasks such as retrieval, generation and so on.

Under the background of NFT-type data retrieval and leveraging the NFT1000 dataset, we introduce a task focused on large-scale, high-similarity image-text retrieval, representing a potential approach in the intersection of AI and blockchain research. This task aims to retrieve target images from a massive collections of highly similar pictures by using tokens' descriptions. Although CLIP models are pre-trained using 400 million image-text pairs from the Internet, their performance on fine-grained classification tasks is somewhat lacking. This indicates that CLIP's training approach

---

[3]As of April 12, 2024, 22:00, the data is sourced from site of https://nftgo.io/macro/market-overview

[4]https://www.nftscan.com/

[5]ERC-721 stands for Ethereum Request for Comment #721. It is a universal NFT standard protocol that defines a series of interfaces for NFT token transactions. For more details, please visit:https://eips.ethereum.org/EIPS/eip-721.

struggles to capture the local semantic information of image-text pairs. To address the limitation, we propose a dynamic masking fine-grained contrastive learning scheme. Through analysis of input images, its dynamic masking module probabilistically masks certain component areas of the image and the corresponding captions. This subtractive approach from the global semantics more fully exposes the local features of the image-text pairs, allowing the model to more specifically align the detailed information of the visual-caption pairs. Our experimental results demonstrate that it is possible to train a model that surpasses the total data's top1 accuracy by 7.4% using only 13% of its training data. This significantly reduces the training overhead and enhances the effectiveness of data utilization.

To quantitatively assess the similarity between a set of images and texts, rather than relying on subjective human judgment, we propose the Comprehensive Variance Index(CVI). It comprehensively considers the similarity within images, captions, and the degree of match between images and texts. Our empirical evidence demonstrates a clear correlation between this index and retrieval accuracy.

**Our main contributions are:** (1) We construct the first NFT visual-text dataset in the field of computer vision. (2) We introduce a task of large-scale, high-similarity image-text retrieval. (3) We design an effective training method for NFT data, using less data but training better models. (4) We propose the Comprehensive Variance Index, a universal metric designed to measure the similarity between images and texts.

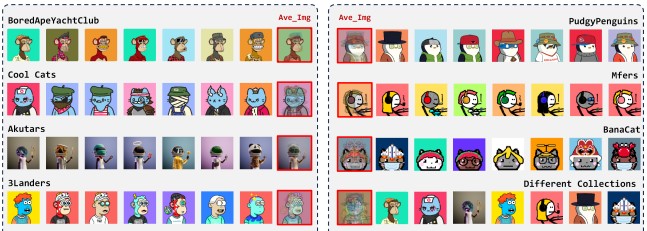

**Figure 2: Randomly selecting seven projects and choosing seven images from each to create an average image (as shown in the red-framed picture), we can observe that the average image has clear contours and distinct content. This indicates that the batch of images randomly selected from the same project possesses a high degree of regional similarity.**

## 2 RELATED WORK

### 2.1 About NFT

NFT [26], short for Non-Fungible Token, is a kind of unique virtual digital asset based on blockchain [29]. As a fundamental component of the metaverse, NFT plays a significant role in various domains, such as social interaction, finance, sports, gaming copyright verification, *etc.* NFT is a broad concept encompassing a diverse array of forms, including images, videos, text, audio, code, and more. Each form of NFT is unique, making them distinct from more common, interchangeable tokens like cryptocurrencies (e.g. Bitcoin [18] and

Ethereum [1]). However, the most widely accepted forms of NFT currently are multimedia formats such as images and videos.

NFT is highly valued due to its unique combination of scarcity, verifiability, liquidity, and the ability to fulfill people's social status needs. NFT is scarce because each one is unique or limited in quantity, making it sought after in a market where people are willing to pay more for rare items. Its possession is verifiable through blockchain technology, which provides a secure, transparent record of each NFT's history and ownership, ensuring authenticity and reducing the risk of fraud. Furthermore, NFT offers high liquidity compared to physical assets; it can be easily bought, sold, or traded on global platforms with minimal transaction costs, making it attractive to investors looking for quick and efficient asset turnover. Lastly, owning an NFT, especially those created by famous artists or those that are particularly rare, can convey social status, as it signifies wealth, taste, and exclusivity. This desire for social recognition through unique digital assets drives demand and increases its value. Collectively, these factors make NFT valuable in today's digital economy, appealing to collectors, investors, and those seeking social distinction alike. According to statistical data[6], prominent NFT projects have achieved significant trading volumes: Bored Ape Yacht Club has sold $3.66 billion, CryptoPunks has amassed $2.78 billion, Mutant Ape Yacht Club has make $2.51 billion, etc.

As the metaverse continues to develop, NFT will increasingly become a digital commodity for trading. As previously mentioned, an NFT can significantly represent the taste of its holder. Therefore, consumers often prefer those that are renowned and align with their personal style. However, with billions of NFT entries, finding one that suits an individual's needs is challenging. Additionally, the high degree of similarity among NFTs adds considerable complexity to their retrieval. Thus, the task of retrieving an NFT is both a critical need and highly challenging, meriting in-depth research and exploration.

### 2.2 Cross-Modal Retrieval

Cross-modal Image-text Retrieval (ITR) is to retrieve the relevant samples from one modality while the queries are expressed in another modality, usually consists of two subtasks: image-to-text (i2t) and text-to-image (t2i). ITR has been witnessed great success in recent years [4, 12, 20] thanks to the rapid development of deep language-vision models [3, 5, 24] and various large-scale multi-modal pre-trained models [8–12, 20, 22]. Most ITR systems deployed in real-world applications are built upon pre-trained models that have been fine-tuned. Generally speaking, the pre-trained models can be divided into two categories according to the their architectures: 1) Fusion-structure models: ALBEF [12] and BLIP [11]. 2) Dual-encoder models: CLIP [20], META-CLIP [8] and ViLEM [2]. The fusion-structure models process text and image inputs simultaneously through a unified network architecture. In these models, image and text data are merged at an early stage and the entire model propagates forward through a single data stream. The drawback of fusion-structure models is their low-efficiency and inflexibility due to the computation of similarity between queries and whole data of another modality during retrieval. While dual-encoder models encode image and text in parallel by independent models and

---

[6]As of April 12, 2024, https://nftgo.io/discover/top-collections

align them by self-supervised contrastive learning. Compared with fusion-structure models, dual-encoder models are more flexible and are much more efficient at zero-shot inference [20]. Dual-encoder models align image and text semantic features into a consistent high-dimensional feature space and the encoders are generally pretrained models. Besides, the computed semantic features of each branch can be stored for fast inferring during retrieval. These advantages make dual-encoder models efficient and flexible to deploy. In this work, we will fine-tune a series of dual-encoder models on our NFT1000 dataset.

### 2.3 Image-Text Dataset

In the realm of computer vision and natural language processing, datasets like Flickr30K [19], COCO [13] and LAION-5B [21] offer vast amount of image-text pairs for diverse applications. Flickr30K is an image-caption dataset widely used in computer vision and natural language processing research. It consists of 31,000 images sourced from the online photo-sharing platform Flickr. Each image in the dataset is paired with five English captions, which provide descriptive annotations written by human annotators. The COCO dataset provides over 200,000 labeled images with detailed instance annotations and The LAION-5B encompasses 5.85 billion CLIP-filtered image-text pairs, making the training of large-scale multi-modal models plausible.

However, Most of the data in the above datasets are collected from the real world, which inherently exhibits significant distributional differences compared to NFT data. In addition to this, images from one NFT project, although different, have fine-grained semantic similarity because they are permutations and combinations of fixed components, as we will discuss in Section3.2, this is a distinctive feature that the aforementioned datasets do not possess. To our knowledge, iCartoonFace benchmark [28] has similar situation with NFT1000, it is a large-scale, high-quality, richly annotated cartoon face recognition dataset, containing 389,678 images of 5,013 cartoon characters. However, this dataset lacks captions corresponding to each image, making it difficult to meet the requirements for cross-modal retrieval.

Given the absence of a dedicated NFT dataset in the computer vision field, in this work, we construct the first benchmark dataset consisting of NFTs, designed to support NFT retrieval and generation tasks.

## 3 PROPERTIES OF NFT1000

### 3.1 Inherent Image-Text Pair Format

Each NFT in the dataset is associated with a metadata resource file, which typically exists in the form of a JavaScript Object Notation (JSON) format. This file uses key-value pairs to describe the attributes of the NFT token(Fig.3).

### 3.2 Fixed-Components Permutation and Combination

The essential reason for the high degree of similarity among NFT images within a same project lies in the fact that all images are permutations and combinations of fixed components. As shown in Fig.4: (a) Images contain a clothing layer named "Navy striped

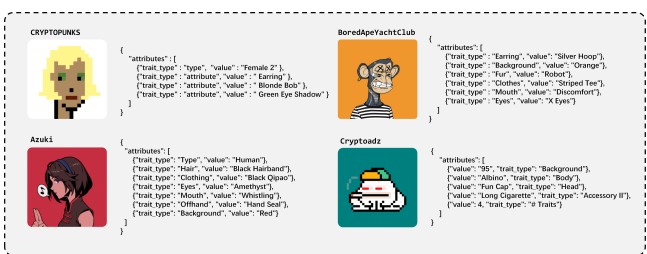

**Figure 3: In the NFT1000 dataset, each image within every collection naturally comes with an accompanying JSON file, which introduces the attributes of the image in a key-value pair format.**

tee"; (b) Pictures include the same "3D glasses" layer. (c) Every image features the same "Bored bubblegum mouth". (d) All photos are adorned with a same "Commie hat". We have selected 20 NFT projects to illustrate the comparison between the number of components and tokens, as shown in the Fig.5. However, it is important to note that in projects initiated after the removal of identical image covers, no two images within a project are the same.

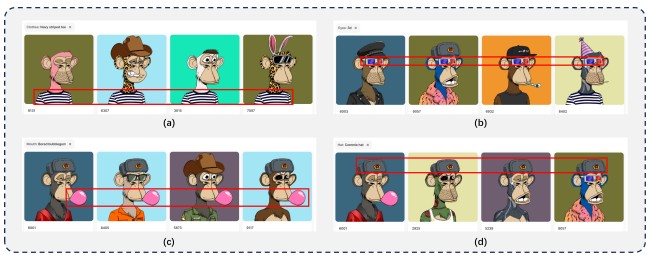

**Figure 4: All images within the same collection are blended from a specific set of components arranged in various combinations, resulting in pixel-level uniformity in image regions.**

### 3.3 Abstract Description

NFT can be considered a form of crypto arts, but the definition of these artworks by artists often include subjective elements. This leads to the abstract description issue, which can be understood as the image itself being difficult to comprehend or the image description lacking clear semantic information. From Fig.6, we can observe intuitively that the No.0 token from the *Superlative Secret Society* project is particularly hard to comprehend, or rather, there is no obvious correlation between its image and caption. It is noteworthy that this situation is common in NFT projects.

## 4 CONSTRUCTING NFT1000

### 4.1 Clarifying the Download Targets

Among various NFT categories, PFP NFT collections account for over 60% of the market share[7]. Besides, in avatar-type NFT, the JSON file accompanying each image relatively effectively describes

---

[7]Please refer to the "Category Market Cap" entry on the Web: https://nftgo.io/analytics/market-overview

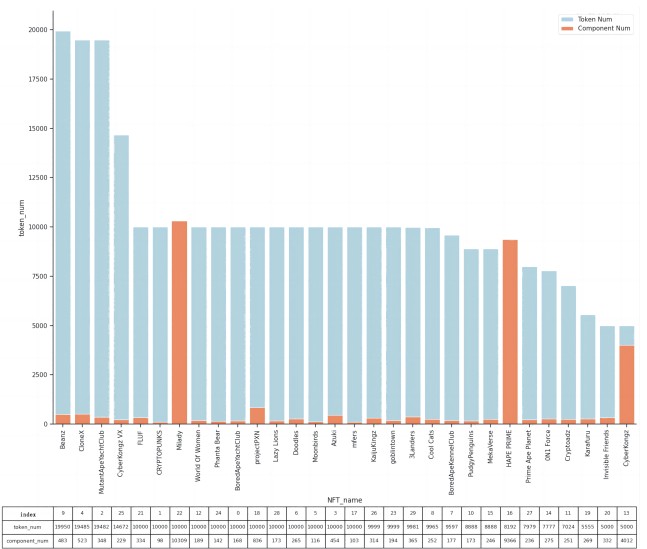

**Figure 5: Illustration of the comparison between the number of components and token in each project..**

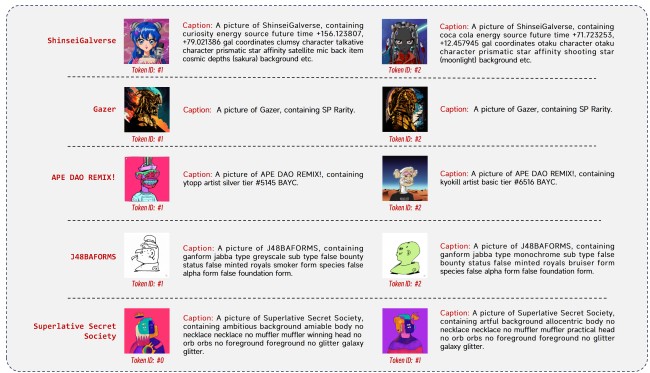

**Figure 6: Show case of abstract images and their abstract descriptions.**

its own attributes. Ethereum is the birthplace of NFT and the most flourishing blockchain for NFT crypto arts. Therefore, we select the top 1000 PFP NFT projects on the Ethereum blockchain, based on sales volume, as our download targets.

### 4.2 Downloading and Filtering

We utilize resources from the Web3.0 domain such as NFTScan[8], Alchemy[9] and IPFS[10], leveraging the basic resource links provided in the smart contracts[11] of each NFT project. This enabled us to piece together the complete links for the media resource and JSON data of each token for downloading and collection. In fact, we have

[8]https://www.nftscan.com/
[9]https://www.alchemy.com/
[10]https://ipfs.tech/
[11]Smart contracts on blockchain are self-executing scripts with the terms written in code.

downloaded resources from a total of 1250 projects for purpose of selection.

Among all the collections that have been fully downloaded, we exclude those with completely duplicated media data (or all images being identical covers), projects with an insufficient total number of tokens (set as fewer than 500), and those lacking a JSON file or where the JSON file contains no substantive semantic information.

### 4.3 Standardization

**Standardize File Format and Dimensions.** Native NFT data, encompassing static image formats such as JPG, PNG, SVG and WebP, are uniformly transformed into the PNG format (This conversion is primarily due to PNG being the predominant format in most NFT collections, and the choice is intended to maximally retain the original fidelity of the data). For dynamic media formats, including GIF and MP4, a representative frame is randomly selected and converted into PNG format. The standardized resolution for these images is set to a width of 512 pixels, with a proportionally adaptive height to maintain aspect ratio integrity. Employing this method, we have reduced the original data size from 14TB to 1.75TB.

**Caption Extraction.** For the original key-value pairs formatted attribute lists, there are two methods for generating captions(Fig.7): one is based on large language models (ChatGPT, LLAMA-13B [23]), using prompt engineering to create descriptions according to the attribute list corresponding to the image; the other way involves using predefined sentence templates to concatenate attributes into a single caption. By using large language models, we generate 30,000 descriptions for 10,000 randomly selected images, while also creating 10,000 captions using language templates. Subsequently, we utilize OpenAI's CLIP-ViT-L pretrained model for zero-shot inference and compared the retrieval accuracy of captions obtained via the two methods (Fig. 8). The result indicates that the large language model can generate better image descriptions, but overall, the performance of the two methods **does not differ significantly.** Lastly, considering the former method would consume considerable time and computational resources, we ultimately opt for generating captions using sentence templates.

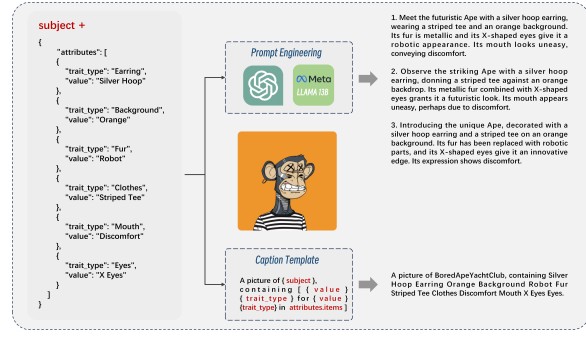

**Figure 7: Illustration of two methods for generating captions**

**Data Partitioning.** Due to the presence of identical components and descriptions in images within the same project, internal division of training and test sets in a NFT collection may result in "data leakage." Consequently, we adopt the project as the fundamental

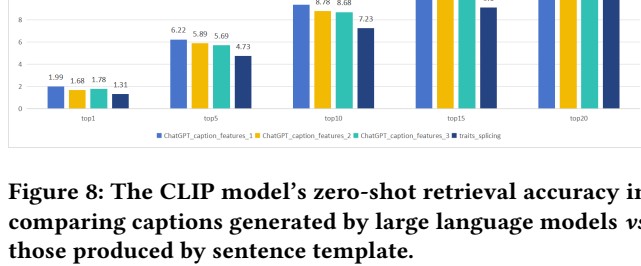

**Figure 8: The CLIP model's zero-shot retrieval accuracy in comparing captions generated by large language models *vs.* those produced by sentence template.**

unit for data division, allocating the entire dataset into training, validation, and test sets in an 80:5:15 ratio.

**Dataset Statistics.** The NFT1000 dataset comprises 1000 outstanding PFP NFT projects, each containing approximately 7500 image-text pairs, encompassing a total of 7.56 million image-text pairs with a collective data volume of 1.75TB. In the dataset, the training set includes 800 projects with 6,178,249 image-text pairs. The validation set comprises 50 projects with 383,916 image-text pairs, and the test set consists of 150 projects with 1,000,838 image-text pairs. The content spans a diverse range of artistic types, including 3D rendered images, 2D flat illustrations, pixel arts, NPC characters, real photographs,etc. It covers a total of 356 different content themes and 595,504 unique descriptive phrases.

## 5 FINE-GRAINED CONTRASTIVE LEARNING

The CLIP models gain fame for achieving state-of-the-art (SOTA) performance through zero-shot inference on various datasets, following its training on a dataset of 400 million image-text pairs using a straightforward contrastive learning strategy. We sequentially use OpenAI's CLIP-ViT-B-32, CLIP-ViT-L-14 pretrained models and META's META-CLIP-ViT-L-14 [8] for zero-shot inference and fine-tuning. The experimental results are presented in Table 1. This table reveals that these pretrained SOTA models have almost never encountered data from the NFT1000 dataset, indicating that the data distribution in NFT1000 is unique and novel. Despite the noticeable improvement (with an average increase in top1 accuracy of about 10%), the overall effectiveness remains suboptimal.

**Table 1: Comparison of zero-shot inference and fine-tuning inference accuracy of different models on the NFT1000 test set.**

| model-type | zero-shot | | | fine-tuning | | |
|---|---|---|---|---|---|---|
| | top1 | top5 | top10 | top1 | top5 | top10 |
| CLIP-VIT-B-32 | 0.01 | 0.02 | 0.03 | 10.63 | 20.32 | 25.19 |
| META-CLIP-VIT-L-14 | 0.00 | 0.01 | 0.02 | 13.06 | 23.68 | 28.81 |
| CLIP-VIT-L-14 | 0.06 | 0.25 | 0.42 | 15.36 | 27.55 | 33.26 |

As discussed in Section 3.2, all images within an NFT project are permutations and combinations of fixed components. Besides, Fig 5 shows that the number of components is relatively small compared to the total number of tokens. Given that the CLIP model is not particularly adept at focusing on the local semantic information of images, we hypothesize that the prerequisite for precise retrieval

is accurate cognition. If we could fine-tune the CLIP model at the component level, it might address the issue of the fine-tuned model not achieving satisfactory recall performance. To verify this hypothesis, we propose a fine-grained fine-tuning strategy based on dynamic masking.

## 5.1 Component Separation

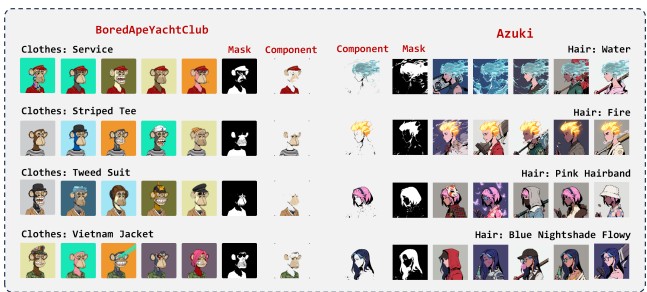

**Figure 9: Illustration of component separation. The results demonstrate that, through a process of initial differentiation followed by superposition, components can be separated into relatively clean and complete entities, even in the presence of overlapping among them.**

Given the pixel-level consistency within the same area of images containing the same component in an NFT project, we adopt a strategy of differentiation followed by superposition to isolate the various distinct components. The specific approach is as follows:

(1) Identify which images share a same component, achievable through analysis of the NFT's accompanying JSON file.
(2) Randomly select a set of images, using the first image as a template, and perform image differencing operations with the subsequent images to get the shared regions and their mask representations.
(3) Repeat step 2 multiple times, ultimately assembling the fragmented components into a relatively complete component and its mask.

Experiments show that performing differencing operations on 4 images at one time and repeating this process 8 times is a good choice. This combination balances execution efficiency and also results in relatively complete and clean components and masks, as shown in Fig. 9.

## 5.2 Dynamic Masking

Before the model loads the training image-text pairs data, we firstly analyze the image to identify its constituent components. With probability $p$, a component's corresponding mask is randomly selected to perform a masking operation on the original image. Simultaneously, the tag of the selected component is removed from the full caption. This process results in a new image-text pair that lacks certain local pixels and descriptive information, thereby allowing the detailed information of the image-text pair to emerge from the global semantics. By subtracting from the original image-text pairs in this manner, the model is encouraged to fully comprehend the correspondence between components and their names, thereby

achieving fine-grained feature alignment with NFT data. A dynamic visualization of the masking process is shown in the Fig. 10.

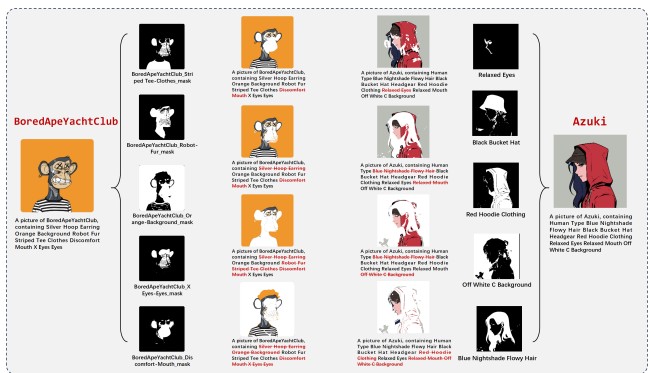

Figure 10: Illustration of the generation process of dynamic masks. Through this method, a single NFT image-text pair can generate new pairs with varied semantic richness.

## 6 EXPERIMENTS ON NFT1000

In this chapter, we will conduct a series of experiments to validate the effectiveness of the dynamic masking fine-tuning method and the application potential of the NFT1000 dataset, focusing on four aspects: the selection of the dynamic masking probability $p$, the generalizability of the dynamic masking approach, the metric Comprehensive Variance Index (CVI) and NFT generation.

In the classical contrastive learning framework, we introduce a dynamic masking unit capable of analyzing the composition of sampled image-text information. This unit applies masks to components of an NFT image with a specific probability, thereby eliminating certain semantic information from the global image-text context. For the remaining training pipeline, we employ the same training strategy as the original CLIP to fine-tune models. Specifically, we utilize image and text encoders to extract features from images and captions. Subsequently, we use contrastive loss to optimize the parameters of the image and text encoders, aiming to progressively align NFT images and their corresponding captions within the same semantic space. The training pipeline is illustrated in Fig. 11.

### 6.1 Mask Selection Probability

During the process of generating dynamic mask, a component mask is selected with a probability $p$. The larger the value of $p$, the more areas of the original image are masked, resulting in finer semantic granularity but also a more fragmented image; conversely, the smaller the value of $p$, the fewer areas are masked, leading to coarser semantic granularity and a more rough correspondence between components and captions. Therefore, selecting an appropriate $p$ is a critical issue.

To swiftly determine the appropriate probability, we construct a smaller dataset from the complete dataset, called NFT1000mini. This subset consists of a training set with 800 projects, a bitch of 1000 image-text pairs are randomly extracted from per project, totaling 794,698 pairs; and a test set comprising 150 projects, each with 1000 random image-text pairs, totaling 147,615 pairs. The comparison of

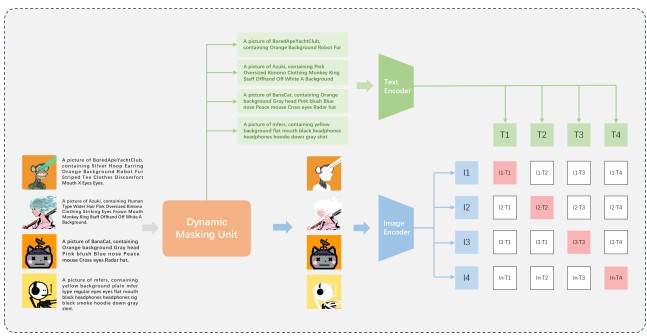

Figure 11: Illustration of the fine-tuning pipeline. The integration of the dynamic masking unit allows for the highlighting of local information within NFT image-text pairs, thereby facilitating fine-grained alignment of the model with NFT data.

Table 2: Comparison of data sizes between NFT1000mini and NFT1000

|  | NFT1000mini | | NFT1000 | |
| --- | --- | --- | --- | --- |
|  | NFT project number | image-text pairs | NFT project number | image-text pairs |
| training set | 800 | 794,698 | 800 | 6,178,249 |
| validation set | 50 | 49, 738 | 50 | 383,916 |
| test set | 150 | 147,615 | 150 | 1,000,838 |

data sizes between NFT1000mini and NFT1000 is shown on Table 2. Subsequently, we conducted a series ablation studies by using the pre-trained CLIP-ViT-B-32 model on the NFT1000mini training set with the same training parameters but varying $p$ for model fine-tuning. With results shown in Table 3 . From the table, we can observe that the relationship between $p$ and accuracy forms a convex function, peaking near $p = 0.5$. This indirectly suggests that the more random the mask selection, the better the training effect of the model. Unless otherwise specified, we set $p = 0.5$ in subsequent experiments.

Table 3: The impact of different mask selection probabilities on model retrieval performance.

| probability | top1 | top5 | top10 |
| --- | --- | --- | --- |
| p=0 | 16.93 | 29.99 | 36.17 |
| p=0.3 | 22.79 | 36.86 | 42.87 |
| p=0.5 | 22.68 | 37.17 | 43.51 |
| p=0.7 | 21.59 | 35.71 | 41.88 |

### 6.2 Generalizability of Dynamic Masking

To verify whether we can fine-tune the model more efficiently under the condition of fine-grained semantic alignment, we compared the inference performance on the NFT1000mini test set of different models trained with and without dynamic masking on the NFT1000mini training set, as well as those trained on the entire NFT1000 training dataset. Subsequently, we obtained surprising results, as shown in Table 4. It is evident that under the same training set conditions (NFT1000mini training set), the use of dynamic masking leads to at least a 10% improvement in accuracy. Compared with the CLIP-ViT-L-14 model, which achieves SOTA performance

using the NFT1000 training set, there's a 7.44% increase in top1 accuracy. This conclusively demonstrates the effectiveness of the dynamic masking training method.

In addition to conducting instance-level searches across the entire dataset, we also compared the search results within a specific NFT project by zero-shot and fine-tuning inference, with data presented in Table 5. It displays the retrieval results for the top 5 and bottom 5 NFT projects, with the data for the top 5 achieving nearly 100% in the top 10 accuracy. However, we can also directly observe that the bottom 5 NFT projects show almost no improvement in accuracy before and after model fine-tuning. This issue arises from the abstract definitions discussed in section 3.3. Consequently, how to retrieve NFT data with abstract definitions will be a focal point of our future work.

## 6.3 Comprehensive Variance Index

To quantitatively measure the similarity between a set of image-text pairs, rather than just relying on subjective human judgment (for example: "not very similar," "somewhat similar," "very similar," *etc.*), we propose the Comprehensive Variance Index. In the current realm of deep learning, a commonly used approach [15] for image-text retrieval involves employing pretrained visual and language encoders to extract image and text features, known as embeddings. Subsequently, dot product operations are conducted to obtain the cosine similarity between the images and texts. The similarity scores are then sorted in descending order to yield the final topk results. For any given model, the most hard retrieval scenario occurs when all probabilities are identical, forcing the model to make a blind selection.

Based on this observation, we propose a concept originating from the probability distribution of vector cosine similarities. This concept posits that if a batch of images exhibits a more uniform distribution of cosine similarity probabilities (in a certain sense, the smaller the variance in the distribution of cosine similarity), the features of these images are more similar. This similarity manifests in semantic and regional aspects of the images, concurrently increasing the difficulty of image retrieval.

Drawing from the preceding discussion, we propose the Comprehensive Variance Index. $I \in \mathbb{R}^{N \times M}$ represents the feature vectors of a batch of images, in which $N$ represents the number of images and $M$ denotes the dimensionality of the feature vectors. Then $S_{II} \in \mathbb{R}^{N \times N}$ is given by $S_{II} = I \cdot I^{\top}$. Similarly, we can obtain the inner product of the corresponding texts' feature vectors, denoted as $S_{TT} \in \mathbb{R}^{N \times N}$, and the inner product of the text-image feature vectors, denoted as $S_{TI} \in \mathbb{R}^{N \times N}$. Following, CVI of a batch of image-text pairs is defined as

$$\begin{aligned} CVI = \frac{1}{2N} \Big( & \alpha \sum_{i=1}^{N} \text{var}(S_{II\_i}) \\ & + (1-\alpha) \sum_{i=1}^{N} \text{var}(S_{TT\_i}) + \sum_{i=1}^{N} \text{var}(S_{TI\_i}) \Big) \end{aligned} \quad (1)$$

where $i$ represents a row in the matrix, $\alpha$ stands for the bias index, indicating the overall metric's preference for the similarity between images and the similarity between captions.

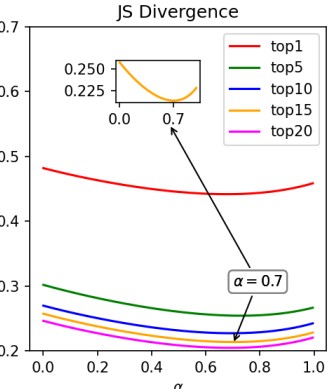

Figure 12: After L1 normalization, the trend of the JSD between the CVI distribution and the TopK distribution varies with changes in alpha. When $\alpha$ approaches 0.7, the JSD approximately reaches its lowest point and CVI can most accurately serves as a measure of image-text similarity.

Jensen-Shannon divergence (JSD) [16] is a popular method for measuring the similarity between two probability distributions. It is a symmetrized and smoothed version of the Kullback-Leibler divergence (KLD) [25]. Given two probability distributions $P$ and $Q$, the JSD is mathematically defined as:

$$JSD(P \parallel Q) = \frac{1}{2}D(P \parallel M) + \frac{1}{2}D(Q \parallel M) \quad (2)$$

where $M = \frac{1}{2}(P + Q)$. One of the key properties of JSD is its boundedness, as it ranges from 0 to 1. A value of 0 indicates that the two distributions are identical, while a value of 1 signifies complete dissimilarity.

Experiments show that when $\alpha$ is approximately 0.7 (Fig.12), CVI best fits the experimental data. This also suggests that the information contained in images is more significant than that in captions and should therefore have a greater weight in similarity measurements. We randomly selected some projects from NFT1000 and some categories from COCO [13] to conduct zero-shot inference using a pretrained CLIP model and to calculate the corresponding CVI values. The results are shown in Table 6, we can see that the lower CVI value, the more similar the batch of image-text pairs is, indicating a higher retrieval difficulty; conversely, a higher CVI value signifies easier retrieval. This also demonstrates that data retrieval within the NFT1000 dataset is indeed a challenging task.

## 6.4 NFT Generation

As discussed in Section 3.1, NFT data inherently comes with a descriptive JSON file, and most NFTs fall within the category of artworks, making them particularly suitable for generative tasks. Leveraging diffusion models [6] and LoRA [7], we trained a LoRA plugin model using image-text pairs from the Azuki NFT project. We then employed image-to-image and text-to-image to create images in the style of Azuki. The results are shown in Fig. 13. Parts a and b involve image-to-image generation based on existing images, while part c involves text-to-image generation to create new styles of images that do not exist in the original project.

Table 4: Inference results of different models on the NFT1000mini test set under various training methods.

| model_type | zero-shot | | | FT-NFT1000mini | | | FT-NFT1000 | | | FT-NFT1000mini-with-dynamic-mask | | |
|---|---|---|---|---|---|---|---|---|---|---|---|---|
| | top1 | top5 | top10 | top1 | top5 | top10 | top1 | top5 | top10 | top1 | top5 | top10 |
| CLIP-VIT-B-32 | 0.03 | 0.10 | 0.15 | 12.10 | 23.66 | 29.54 | 20.33 | 34.47 | 40.74 | 22.68 ↑2.35 | 37.17 ↑2.7 | 43.51 ↑2.77 |
| META-CLIP-VIT-L-14 | 0.01 | 0.05 | 0.11 | 20.53 | 35.05 | 41.67 | 23.07 | 37.08 | 43.01 | 31.83 ↑8.76 | 47.29 ↑10.21 | 53.47 ↑10.46 |
| CLIP-VIT-L-14 | 0.33 | 1.02 | 1.55 | 20.43 | 34.78 | 41.13 | 26.66 | 41.91 | 48.22 | 34.10 ↑7.44 | 50.21 ↑8.3 | 56.41 ↑8.19 |

Table 5: The recall rate within NFT project before and after fine-tuning the CLIP-ViT-L-14 model.

| collection | item_num | zero-shot | | | fine-tuning | | |
|---|---|---|---|---|---|---|---|
| | | top1 | top5 | top10 | top1 | top5 | top10 |
| Stoner Ape Club | 6666 | 1.08 | 2.81 | 4.29 | 91.31 | 98.93 | 99.61 |
| Junglebayapeclub | 5555 | 0.90 | 2.65 | 4.14 | 89.79 | 98.56 | 99.23 |
| Cool Ape Club | 5555 | 0.59 | 1.39 | 2.32 | 88.17 | 97.95 | 99.05 |
| Fat Rat Mafia | 7777 | 0.03 | 0.27 | 0.63 | 83.27 | 96.18 | 98.06 |
| 0xAzuki | 9999 | 0.85 | 3.25 | 5.48 | 80.73 | 95.44 | 97.82 |
| …… | …… | …… | …… | …… | …… | …… | …… |
| ShinseiGalverse | 8889 | 0.01 | 0.16 | 0.35 | 0.19 | 0.75 | 1.31 |
| Gazer | 2100 | 0.00 | 0.24 | 0.43 | 0.05 | 0.19 | 0.43 |
| APE DAO REMIX! | 5528 | 0.02 | 0.14 | 0.22 | 0.04 | 0.18 | 0.36 |
| J48BAFORMS | 4848 | 0.04 | 0.10 | 0.21 | 0.12 | 0.27 | 0.54 |
| Superlative Secret Society | 11110 | 0.02 | 0.05 | 0.08 | 0.02 | 0.08 | 0.21 |
| all_collections | 1000838 | 0.06 | 0.25 | 0.42 | 21.04 | 34.40 | 40.16 |

Table 6: Comparison table of retrieval accuracy and CVI between NFT1000 and COCO datasets.

| | Collection/Category | Top1 | Top5 | Top10 | CVI |
|---|---|---|---|---|---|
| NFT1000 | Savage Droids | 0.0365 | 0.1823 | 0.3646 | 0.0003 |
| | Hor1zon | 0.0286 | 0.1429 | 0.4858 | 0.0005 |
| | CyberTurtles | 0.3060 | 0.7201 | 1.3141 | 0.0007 |
| | SpriteClub | 0.4758 | 1.7359 | 2.9574 | 0.0009 |
| | Tasty Bones | 1.4656 | 3.7433 | 5.9418 | 0.0011 |
| COCO | person | 10.3192 | 25.6125 | 37.2309 | 0.0034 |
| | car | 13.2463 | 36.3806 | 50.0000 | 0.0037 |
| | broccoli | 18.0556 | 37.5000 | 56.9444 | 0.0038 |
| | backpack | 24.8908 | 52.8384 | 68.1223 | 0.0042 |
| | cell phone | 26.0465 | 50.2326 | 60.4651 | 0.0044 |

This illustration demonstrates that downstream models generated by pre-trained diffusion models and LoRA can accurately capture the stylistic features of a specific NFT project. Furthermore, images generated from the same set of prompts exhibit high coherence and aesthetic appeal, which are crucial for NFT collections.

## 7 DISCUSSION AND FUTURE WORK

This section will discuss potential improvements methods and future work for this study.

### 7.1 Efficient Utilization of Data

As demonstrated by the Table 4, by utilizing only 13% of the NFT1000 training dataset, we have successfully trained a superior model, suggesting a potential redundancy within NFT-type data. This prompts a question: What is the minimum amount of data required to maintain experimental accuracy? Efficient data utilization remains an area for exploration. Moreover, the issue of effectively retrieving NFT projects with abstract definitions, as discussed in Section 3.3, also warrants further investigation.

### 7.2 Continuing to Expand the Dataset

NFT1000 is an ambitious project. In the future, we plan to broaden our scope beyond Ethereum to include more collections of outstanding NFTs from other public blockchains like Solana, Polygon, BNB Chain, Klaytn, etc. We aim to scale the data to the level of hundreds of millions, striving to build an ImageNet equivalent in the NFT domain, thereby making a significant contribution to both the academic and industrial communities.

### 7.3 Exploring Further Potential of NFT1000

NFT holds significant untapped potential for development. In the future, we plan to explore the use of generative models to create a wider array of NFT artworks.

## 8 CONCLUSION

In this work, we construct the first NFT visual-text dataset in the field of computer vision. and introduce a task for large-scale, high-similarity image-text retrieval. Furthermore, we propose an effective training method for NFT-type data, called dynamic masking fine-tuning scheme, and have trained several models as our baseline. To quantify image-text similarity, we introduce the Comprehensive Variance Index, which accounts for the similarities within images and texts, as well as the degree of image-text matching. Finally, we also explore the application of NFT data in the image generation field, paving a feasible path for future AI-generated content creation (AIGC) for NFTs.

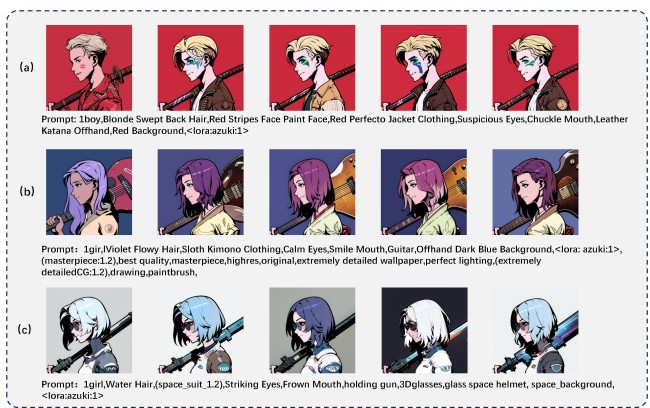

Figure 13: Effect of NFT generation based on diffusion models.

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
