# OpenReview forum: "NFT1000: A Cross-Modal Dataset For Non-Fungible Token Retrieval"
_acmmm.org/ACMMM/2024/Conference — MM2024 Poster_

### Official Review · Reviewer_8fW8 · 2024-05-19

**Rating:** 5
**Confidence:** 2

**Summary:**

This paper builds and introduces a benchmark dataset named “NFT Top1000 Visual-Text Dataset” for NFT retrieval. Based on the dataset, it also proposes the dynamic masking fine-tuning scheme and the robust metric Comprehensive Variance Index.

**Strengths:**

1. The paper is clearly written and includes a variety of figures and tables.
2. The contributions are substantial, This paper constructs the first NFT visual-text dataset in the field of computer vision and defines a new cross-modal retrieval task based on this dataset, proposing corresponding methods and evaluation metrics.
3. The innovative approach results in a notable performance improvement.

**Limitations:**

I'm not very knowledgeable about NFT retrieval, but as a dataset construction-oriented paper, I believe there is no significant limitations.
Some improvement suggestions are as follows:
1. It is recommended that adding a figure to provide a comprehensive description of the dataset's composition and content.
2. The arrangement of figures and tables needs improvement. For instance, it is recommended that Figure 13 should not appear after the conclusion.

**Suitability:**

3

---

### Official Review · Reviewer_RyYN · 2024-05-23

**Rating:** 2
**Confidence:** 2

**Summary:**

This paper presents a novel NFT-related multimodal dataset and provides an analysis that could be beneficial for future research in the field. It aims to address the classification of NFT images using deep learning models, with a particular focus on enhancing performance through preprocessing steps such as noise removal and image compression. The proposed methodology has potential implications for the broader context of metaverse and Web 3.0 technologies.

**Strengths:**

1. The introduction of a new NFT-based multimodal dataset is a significant contribution, providing valuable resources for the academic and practical exploration of NFTs.

2. The analysis provided in the paper offers insights into the dataset's utility and relevance, which could be instrumental for further research in this emerging domain.

**Limitations:**

1. The motivation for analyzing multimodal content on NFTs is not clearly articulated in the introduction and dataset description sections. The paper mentions the potential value in metaverse and Web 3.0 contexts but fails to elaborate on specific benefits or the impact on methodological design and subsequent applications. This lack of clarity may alienate readers who are not already engaged with NFT research, hindering the understanding of the dataset's practical value. A more detailed explanation is recommended.

2. The paper provides a link to the demo and dataset, which is currently inaccessible. Ensuring that resources are readily available is crucial for transparency and reproducibility in research. It is recommended to check and update the dataset link.

3. While the dataset itself is a valuable addition, the analysis performed in the study appears limited. Expanding on the experiments and offering new insights based on the dataset could significantly enhance the paper's impact.

4. The potential applications of the dataset in other scenarios or downstream tasks are not sufficiently explored. It would be beneficial to dedicate more discussion to how this dataset can be utilized in various research and practical contexts beyond the scope of the current analysis.

**Suitability:**

2

---

### Official Review · Reviewer_FjBx · 2024-05-24

**Rating:** 3
**Confidence:** 3

**Summary:**

The paper addresses the issue of high similarity in NFT retrieval by constructing, for the first time, a visual-text dataset specifically designed for the field of computer vision, named "NFT1000". This dataset covers the top 1000 PFP NFT projects on the Ethereum mainnet, containing 75 million image-text pairs. Based on this dataset, researchers introduced a dynamic masking fine-tuning scheme to enhance the performance of the existing CLIP model in fine-grained image-text classification tasks. The paper also introduces a new metric, the Composite Variation Index (CVI), to quantify the similarity and retrieval difficulty between images and texts.

**Strengths:**

1. **Innovative dataset construction**: The NFT1000 dataset provides new data resources for the field of computer vision.
2. **Methodological innovation**: The dynamic masking fine-tuning scheme offers a new approach to address the retrieval issues of highly similar NFTs, improving model performance in specific tasks.
3. **Practical application value**: The proposed CVI metric and improved retrieval methods can be practically applied in the NFT market, helping users search and filter NFTs more effectively, with strong market application prospects.

**Limitations:**

1. **Insufficient research motivation**: Each NFT is identified by its unique identifier, and considering the minor differences in data structure and appearance among different NFTs, users may more likely use this unique identifier for retrieval and transactions. In such cases, the motivation for the study needs further justification, especially clarifying why complex image and text retrieval techniques are still needed even with unique identifiers, particularly in large-scale, high-similarity NFT datasets.
2. **Lack of technical detail**: The paper lacks detailed descriptions of the dynamic masking technique and adjustments to the CLIP model, which may affect other researchers' ability to replicate and verify results.
3. **Limited experimental design**: Although the paper demonstrates the model's performance on the NFT1000 dataset, it lacks extensive comparative testing with other datasets or existing methods, limiting the validation of the method's universality.
4. **Lack of analysis of long-term impact**: The paper does not thoroughly explore the long-term impacts of its methods, such as the economic effects on the NFT market and legal and ethical issues.
5. **Severe issues in writing and readability**: For example, in the abstract, the description of "7.4% improvement in the top1 accuracy rate" lacks background information, such as what the improvement is compared to and what the actual significance of this improvement is; although it mentions the "dynamic masking fine-tuning scheme", it does not briefly explain the basic principle of this method or its difference from existing methods; and there are significant readability issues with the images, where the text is completely unreadable.

and other questions：

1. **Scalability and generalizability**: It remains to be validated whether the methods proposed in the paper are applicable to other types of NFTs or more broadly to image-text datasets.
2. **Data security and privacy concerns**: The paper does not discuss data security and privacy protection issues involved in collecting and processing NFT data, which are particularly important in the digital asset field.
3. **Economic and social impacts**: As an emerging digital asset, NFTs have complex and variable economic and social impacts. The paper should consider the potential impacts of its research findings on the entire NFT market and stakeholders such as artists and collectors.

**Suitability:**

2

---

### Official Review · Reviewer_ixoa · 2024-05-24

**Rating:** 6
**Confidence:** 3

**Summary:**

In this paper, the authors introduce a non-fungible token (NFT) dataset. It contains 7.56 million image-text pairs. Based on this dataset, the authors also propose a dynamic masking fine-tuning scheme, which results in a 7.4% improvement in terms of top-2 accuracy rate, while utilizing merely 13% of the total training data. At the same time, the authors propose a robust metric comprehensive variance index to assess the similarity and retrieval difficulty of visual-text pairs data.

**Strengths:**

1.	This paper presents a non-fungible token dataset, which is useful for related tasks. Especially, it is a large dataset containing 7.56 million image-text pairs collected from 1000 most famous PFP NFT collections.
2.	A dynamic masking fine-tuning scheme is proposed, which is effective to improve the performance. This approach can be used as a baseline for related tasks.
3.	The paper is well organized and written.

**Limitations:**

1.	There are several minor writing problems, e.g., grammars, sentence organization.

**Suitability:**

3

---

### Meta-Review · Area_Chair_BJcB · 2024-07-01

**Recommendation:** Accept (Poster)
**Confidence:** 4

**Metareview:**

Summary:

This paper introduces the NFT1000 dataset, a large collection of image-text pairs from the top 1000 PFP NFT collections. It proposes a dynamic masking fine-tuning scheme that significantly improves retrieval performance and introduces a new metric named Composite Variation Index (CVI) to assess similarity and retrieval difficulty.

Strengths:
1. The paper presents a large and valuable NFT dataset, providing new resources for multimedia research.
2. The dynamic masking fine-tuning scheme improves retrieval performance, offering a potential baseline for related tasks.
3. The CVI metric and enhanced retrieval methods have strong practical applications in the NFT market, aiding in effective search and filtering.

Limitations:
1. The paper needs clearer justification for complex retrieval techniques given the unique identifiers of NFTs.
2. Detailed descriptions of the dynamic masking technique and adjustments to the CLIP model are missing, affecting replicability.
3. The paper lacks extensive comparative testing with other datasets or existing methods, limiting the validation of the method.
4. There are several minor writing problems, such as grammar and sentence organization.

According to the reviewers' feedback after the rebuttal, two reviewers who lean to reject the submission raised their rating, so the overall rating of this paper is positive. I tend to recommend this paper as Accept (Poster).